# METHODOLOGY FOR THE COMPREHENSIVE STUDY OF A MULTIDIMENSIONAL STATISTICAL SAMPLE IN THE DIAGNOSIS OF SPINAL DISEASES

## Abstract

The report proposes a methodology for the analysis of statistical samples of multidimensional data. According to this methodology, the centers of the clusters in the studied statistical sample are determined through comprehensive application of cluster analysis methods. These cluster centers are associated with values of neurogrowth metrics, Euclidean metric, and a combined metric defined as the product of the neurogrowth metric and the Euclidean metric. The methodology is demonstrated on a statistical sample used in the diagnosis of spinal diseases. The task is to identify which factors (intervertebral distances in the lumbar spine: $z_1$, $z_2$, $z_3$, $z_4$) most influence the transition of patients and their diagnoses from one cluster to another. Sensitivity analysis of the metrics to variations in the intervertebral distances $z_1$, $z_2$, $z_3$, $z_4$ was performed, showing a high sensitivity to changes in $z_2$ (probability 100%), substantial sensitivity to $z_1$ (88%), and low sensitivity to $z_3$ (60%). This methodology enables the identification of factors responsible for transitions between patient clusters, thereby contributing to improved early diagnosis methods for osteochondrosis.

Keywords: lumbar spine, diagnosis of osteochondrosis, multidimensional data, cluster analysis, neural metric, sensitivity analysis, decision formalization

## 1 Introduction

To determine the signs of the presence or absence of "osteochondrosis" of the lumbar spine, measurements of 4 intervertebral distances of polyclinic patients were considered. The distances were measured from preventive X-ray images (Fig. 1) by a polyclinic specialist, who also indicated control points – a patient with the presence and a patient with the absence of "osteochondrosis".Barsegyan et al. (2004)

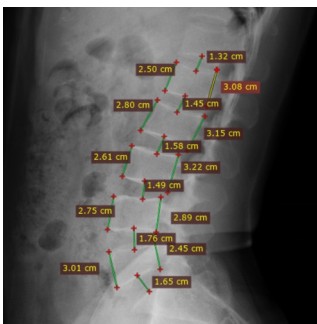

Figure 1: Measurements of distances between vertebrae in the lumbar spine

Using cluster analysis methods Yakimov et al. (2011), non-overlapping sets of patients (3 clusters) were obtained – with presence, with absence of "osteochondrosis", and borderline cases. Cluster centers were determined.

Table 1 presents the designation of the centers of 3 clusters with the presence of pathology, absence of pathology, and borderline cases. It is necessary to recognize these diagnoses based on the data from Table 1 and determine changes in which factors (intervertebral distances $Z_1, Z_2, Z_3, Z_4$) have the greatest influence on the transition of patients and their diagnoses from one cluster to other clusters.

Table 1: Cluster centers

| Cluster | $Z_1$ (mm) | $Z_2$ (mm) | $Z_3$ (mm) | $Z_4$ (mm) |
|---|---|---|---|---|
| Healthy | 12.5 ± 1.2 | 14.3 ± 0.8 | 11.2 ± 0.9 | 13.1 ± 1.1 |
| Pathology | 8.7 ± 1.5 | 9.1 ± 1.3 | 7.8 ± 1.4 | 8.9 ± 1.2 |
| Borderline | 10.6 ± 1.3 | 11.8 ± 1.0 | 9.5 ± 1.1 | 10.9 ± 1.0 |

Elements $z_{ij} \in R$, $z_{ij} \in [0,1]$, $i = 1, \ldots, 3$, $j = 1, \ldots, 4$.

## 2 Analysis of multidimensional X-ray data of the lumbar spine

### 2.1 Qualitative and quantitative characteristics of clusters

Description of the patient sample, X-ray images, measured parameters (intervertebral distances $z_1$, $z_2$, $z_3$, $z_4$). Let us introduce notation for cluster centers

$C_i = (z_{i1}, z_{i2}, z_{i3}, z_{i4})$, $i = 1, \ldots, 3$.

Let us associate with the cluster centers presented in the rows of Table 1 the values of the neuron growth metric Neur_M, Euclidean metric Evcl_M, and the metric Metric(Neur_M, Evcl_M) representing the product of the values Neur_M and Evcl_M

$$\text{Neur\_M}(C_i) = \sum_{j=1}^{4} z_{ij} w_j, \quad w_j \in \mathbb{R}, \quad i = 1, \ldots, 3, \quad j = 1, \ldots, 4. \tag{1}$$

$$\text{Evcl\_M}(C_i) = \sqrt{\sum_{j=1}^{4} z_{ij}^2} \tag{2}$$

$$\text{Metric}(C_i) = \text{Evcl\_M}(C_i) \cdot \text{Neur\_M}(C_i) = \sqrt{\sum_{j=1}^{4} z_{ij}^2} \cdot \sum_{j=1}^{4} z_{ij} w_j. \tag{3}$$

Let us consider various cases of selecting parameters $w_1$, $w_2$, $w_3$, $w_4$ of the neuron growth (1), satisfying the conditions

$$\sum_{j=1}^{4} w_j = 0 \tag{4}$$

$$w_{j-1} \leq w_j, \quad j = 2, \ldots, 4. \tag{5}$$

In this case:

1) the value Neur_M$(C_i) < 0$ corresponds to pathology (values $z_{i1}, z_{i2}, z_{i3}, z_{i4}$ do not increase),

2) the value Neur_M$(C_i) = 0$ corresponds to the case $z_{i1} = z_{i2} = z_{i3} = z_{i4}$,

3) the value Neur_M($C_i$) >0 corresponds to an increase in the values $z_i1$, $z_i2$, $z_i3$, $z_i4$.

The values of the Metric($C_i$) for cluster centers data from Table 1 are analogous to the area values of rectangles with sides Neur_M, Evcl_M. A value Metric($C_i$) < 0 corresponds to a cluster with the presence of pathology. The maximum value of Metric corresponds to the group of healthy patients. The average of the 3 Metric values (3) for clusters corresponds to the group of borderline cases excluding the exact presence or absence of pathology.

Let us perform additional studies of the considered data. It is necessary to establish the range of variation of metrics (1), (2), characterizing the clusters of each component of the parameter vector $C = (z_1, z_2, z_3, z_4)$. In accordance with the range of variation of responses (1), (2), the decision-making strategy regarding the diagnosis of osteochondrosis is determined. If with a significant amplitude of change of some component of the parameter vector $C = (z_1, z_2, z_3, z_4)$ the responses (1), (2) change insignificantly, this means that the accuracy of representing this intervertebral distance does not play a significant role. Furthermore, in planning the conclusion or treatment procedures, this intervertebral distance will not be used as the main one. If the responses (1), (2) turn out to be highly sensitive to changes in some component of the vector $C = (z_1, z_2, z_3, z_4)$, then this serves as a direct indication of the need to represent it in the diagnostic model with the highest priority Максимей (1988).

## 3 Influence of weight coefficients and intervertebral distances on cluster characteristics

### 3.1 Case 1. Small weights

Let the neuron growth Neur_M (1) have the following weight coefficients $w_1 = -0.6, w_2 = 0.1, w_3 = 0.2, w_4 = 0.3$.

Table 2 presents the previously determined centers of 3 patient clusters and the corresponding values of metrics (1) – (3).

Table 2: Example of a filled table with cluster centers and metric values (1) – (3) for $w_1 = -0.6, w_2 = 0.1, w_3 = 0.2, w_4 = 0.3$

| $Z_1$ | $Z_2$ | $Z_3$ | $Z_4$ | Neur_M | Evcl_M | Pathology | Metric |
|-------|-------|-------|-------|--------|--------|-----------|--------|
| 0.73 | 0.93 | 0.62 | 0.64 | -0.03 | 1.48 | + | -0.04 |
| 0.60 | 0.64 | 0.75 | 0.72 | 0.07 | 1.36 | ± | 0.10 |
| 0.80 | 0.84 | 0.94 | 0.92 | 0.07 | 1.75 | - | 0.12 |

According to observations, when Metric($C_i$) <0, there is pathology (osteochondrosis of the lumbar spine), the maximum value of Metric($C_i$) (3) corresponds to the absence of pathology, the average value (3) of the 3 values of this metric for rows of clusters 1-3 corresponds to borderline cases regarding the presence of diagnosis (see Fig. 2.).

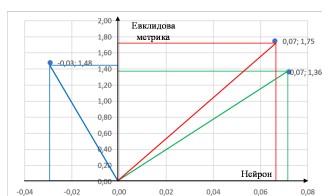

Figure 2: Geometric interpretation of the Metric($C_i$) for 3 clusters of data from Table 2 – areas of rectangles with sides Neur_M, Evcl_M.

A sensitivity analysis (Максимей, 1988) of changes in the Neuron Neur_M (1) and Evcl_M (2) metrics to changes in the values of components $z_{ij}$, $i = 1, \ldots, 3$, $j = 1, \ldots, 4$ at 2 levels for the cases indicated in Fig. 3 was performed.

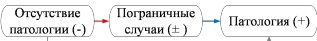

Figure 3: Transition of patients between clusters

The ratios of changes in the Neuron metric Neur_M (1) with changes in the corresponding components $z_i$, the ratios of changes in the Euclidean metric Evcl_M (2) with changes in the corresponding components $z_i$ were determined as percentages, and the values of the Euclidean metric from the pairs of ratios described above and presented in the graphs in Fig. 3 were determined.

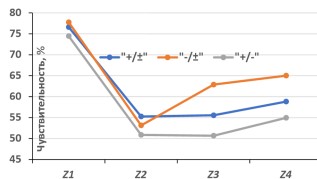

Figure 4: Results of sensitivity analysis of metrics Neur_M and Evcl_M to changes in the values of factors $z_{ij}$ during transitions between clusters indicated in Fig. 2.

Table 3: Sensitivities of Neur_M and Evcl_M to changes in components $z_i$ during transition from the cluster with pathology to borderline cases

| Parameter | $N$ | $E$ | $E(N, E)$ |
|-----------|-----|-----|-----------|
| $z_1$ | 60 | 48 | 77 |
| $z_2$ | 10 | 54 | 55 |
| $z_3$ | 20 | 52 | 56 |
| $z_4$ | 30 | 51 | 59 |

Analysis of the data from graphs in Fig. 3 and tables in Table 3 - Table 5 allows ranking the responses (1), (2) by (decreasing) degree of sensitivity to changes in components $z_i$.

The metric Neur_M – $N$ in tables 3, 4, 5 – for given parameters $w_i$, $i = 1, \ldots, 4$, is not sensitive to changes in the values of components z2, z3 during patient transitions between clusters.

The sensitivity of the Euclidean metric – $E$ in tables 3, 4, 5 – by degree of decrease to changes in components $z_i$ corresponds to the order of significance of changes in the values of components $z_i : z_1, z_4, z3, z_2$.

The joint sensitivity of the Euclidean metric and the neuron metric defined by the Euclidean – $E(N,E)$ in tables 3-5 defined by the Euclidean metric between the values of $N$ and $E$ in these tables – by degree of decrease to changes in components $z_i$ corresponds to the order of significance of changes in the values of components $z_i : z_1, z_4, z3, z_2$, which is due to the greater influence of the values of parameters $w_i$ in (1) on the analysis results.

### 3.2 Case 2. Large weights

Let the values of the weight coefficients of the "Neuron growth" metric Neur_M (1) be $w_1 = -6, w_2 = 1, w_3 = 2, w_4 = 3$.

Table 6 presents the values (1) – (3) corresponding to the patient cluster centers.

The results of the sensitivity analysis of responses Neur_M and Evcl_M to changes in components $z_i$, $i = 1, \ldots, 4$, during patient transitions between clusters indicated in Fig. 2 are shown in Figure 4 and in tables 6-8.

The joint sensitivity of the Euclidean metric and the neuron metric defined by the Euclidean – $E(N,E)$ in tables 6-8 defined by the Euclidean metric between the values of $N$ and $E$ in

Table 4: Sensitivities of Neur_M and Evcl_M to changes in components $z_i$ during transition from the cluster without pathology to borderline cases

| Parameter | $N$ | $E$ | $E(N, E)$ |
|---|---|---|---|
| $z_1$ | 60 | 49 | 78 |
| $z_2$ | 10 | 52 | 53 |
| $z_3$ | 20 | 60 | 63 |
| $z_4$ | 30 | 58 | 65 |

Table 5: Sensitivities of Neur_M and Evcl_M to changes in components $z_i$ during transition from the cluster with pathology to the cluster without pathology

| Parameter | $N$ | $E$ | $E(N, E)$ |
|---|---|---|---|
| $z_1$ | 60 | 44 | 74 |
| $z_2$ | 10 | 50 | 51 |
| $z_3$ | 20 | 47 | 51 |
| $z_4$ | 30 | 46 | 55 |

Table 6: Filled table with cluster centers and metric values $(1) - (3)$ for $w_1 = -6,\quad w_2 = 1,\quad w_3 = 2,\quad w_4 = 3$

| $Z_1$ | $Z_2$ | $Z_3$ | $Z_4$ | Neur_M | Evcl_M | Pathology | Metric |
|---|---|---|---|---|---|---|---|
| 0.73 | 0.93 | 0.62 | 0.64 | -0.29 | 1.48 | $+$ | -0.43 |
| 0.60 | 0.64 | 0.75 | 0.72 | 0.72 | 1.36 | $\pm$ | 0.98 |
| 0.80 | 0.84 | 0.94 | 0.92 | 0.66 | 1.75 | $-$ | 1.16 |

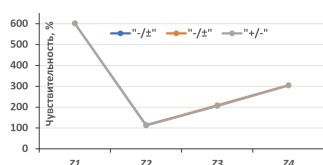

Figure 5: Results of sensitivity analysis of metrics Neur_M and Evcl_M to changes in the values of factors $z_{ij}$ during transitions between clusters indicated in Fig. 3.

Table 7: Sensitivities of Neur_M and Evcl_M to changes in components $z_i$ during transition from the cluster with pathology to borderline cases

| Parameter | $N$ | $E$ | $E(N, E)$ |
|---|---|---|---|
| $z_1$ | 600 | 48 | 602 |
| $z_2$ | 100 | 54 | 114 |
| $z_3$ | 200 | 52 | 207 |
| $z_4$ | 300 | 51 | 304 |

Table 8: Sensitivities of Neur_M and Evcl_M to changes in components $z_i$ during transition from the cluster without pathology to borderline cases

| Parameter | $N$ | $E$ | $E(N, E)$ |
|---|---|---|---|
| $z_1$ | 600 | 49 | 602 |
| $z_2$ | 100 | 52 | 113 |
| $z_3$ | 200 | 60 | 209 |
| $z_4$ | 300 | 58 | 305 |

Table 9: Sensitivities of Neur_M and Evcl_M to changes in components $z_i$ during transition from the cluster with pathology to the cluster without pathology

| Parameter | $N$ | $E$ | $E(N, E)$ |
|-----------|-----|-----|-----------|
| $z_1$ | 600 | 44 | 602 |
| $z_2$ | 100 | 50 | 112 |
| $z_3$ | 200 | 47 | 205 |
| $z_4$ | 300 | 46 | 304 |

these tables – by degree of decrease to changes in components $z_i$ corresponds to the order of significance of changes in the values of components $z_i : z_1, z_4, z_3, z_2$, which is due to the greater influence on the analysis results of the values of parameters w1, w2, w3, w4 of the neuron metric (1). The values of $N$ in tables 6-8 exceed 100%, while the values $0 \leq E \leq 100$. The graphs in Fig. 4 are indistinguishable.

A reduction in the values of parameters $w_i$, $i = 1, \ldots, 4$ to values of responses Neur_M($C_i$) (1) proportional to the values of Evcl_M (2) is proposed. For example, the values $w_i$, $i = 1, \ldots, 4$, from case 1 – reduced by 10 times values of $w_i$ from case 2.

## 3.3 Case 3. Normalized weights

Let us normalize the values $w_i$, $i = 1, \ldots, 4$, from case 2. We obtain $w_1 = -1$, $w_2 = 0.17$, $w_3 = 0.33$, $w_4 = 0.5$. Table 9 presents the values of metrics (1) – (3) corresponding to the patient cluster centers.

Table 10: Filled table with cluster centers and metric values (1) – (3) for $w_1 = -1$, $w_2 = 0.17$, $w_3 = 0.33$, $w_4 = 0.5$

| $Z_1$ | $Z_2$ | $Z_3$ | $Z_4$ | Neur_M | Evcl_M | Pathology | Metric |
|-------|-------|-------|-------|--------|--------|-----------|--------|
| 0.73 | 0.93 | 0.62 | 0.64 | -0.05 | 1.48 | $+$ | -0.07 |
| 0.60 | 0.64 | 0.75 | 0.72 | 0.12 | 1.36 | $\pm$ | 0.16 |
| 0.80 | 0.84 | 0.94 | 0.92 | 0.11 | 1.75 | $-$ | 0.19 |

The results of the sensitivity analysis of responses Neur_M and Evcl_M to changes in components $z_i$, $i = 1, \ldots, 4$, during patient transitions between clusters indicated in Fig. 2 are shown in Figure 5 and in tables 10-12.

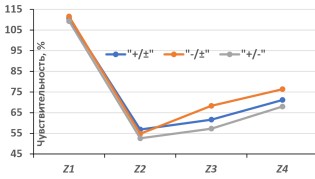

Figure 6: Results of sensitivity analysis of metrics Neur_M and Evcl_M to changes in the values of factors $z_{ij}$ during transitions between clusters indicated in Fig. 3.

The joint sensitivity of the Euclidean metric and the neuron metric, defined by the value $E(N, E)$ in tables 11–13, by degree of decrease to changes in components $z_i$ corresponds to the order of significance of their changes: $z_1, z_4, z_3, z_2$. This is due to the greater influence on the analysis results of the weight coefficients $w_1, w_2, w_3, w_4$ of the neuron metric (1). The values of $N$ and $E$ in tables 6–8 are in the ranges $0 \leq N \leq 100$ and $0 \leq E \leq 100$. However, with parameters $w_1 = -1, w_2 = 0.17, w_3 = 0.33, w_4 = 0.5$, cases of $E(N, E) > 100$ are observed. In this regard, a reduction of the given parameters $w_i$ ($i = 1, \ldots, 4$) is proposed.

Table 11: Sensitivities of Neur_M and Evcl_M to changes in components $z_i$ during transition from the cluster with pathology to borderline cases

| Parameter | $N$ | $E$ | $E(N, E)$ |
|---|---|---|---|
| $z_1$ | 100 | 48 | 111 |
| $z_2$ | 17 | 54 | 57 |
| $z_3$ | 33 | 52 | 62 |
| $z_4$ | 50 | 51 | 71 |

Table 12: Sensitivities of Neur_M and Evcl_M to changes in components $z_i$ during transition from the cluster without pathology to borderline cases

| Parameter | $N$ | $E$ | $E(N, E)$ |
|---|---|---|---|
| $z_1$ | 100 | 49 | 112 |
| $z_2$ | 17 | 52 | 54 |
| $z_3$ | 33 | 60 | 68 |
| $z_4$ | 50 | 58 | 76 |

## 3.4  Case 4. Equivalently transformed weights

Let us reduce the values $w_i$, $i = 1, \ldots, 4$, from case 2 by 100 times. We obtain $w_1 = -0.10$, $w_2 = 0.02$, $w_3 = 0.03$, $w_4 = 0.05$. Table 13 presents the values of metrics (1) – (3) corresponding to the patient cluster centers.

For given parameters $w_1 = -0.10, w_2 = 0.02, w_3 = 0.03, w_4 = 0.05$ and rounding to 2 hundredths in the table cells, the neuron (1) for cluster 1 with presence of pathology takes the value Neur_M = 0 instead of Neur_M < 0 – does not detect pathology. For clusters with absence of pathology and borderline cases, the neuron (1) takes the same value Neur_M = 0.01. Thus, according to the metrics Neur_M and Metric, clusters with absence of pathology and borderline cases are perceived as one cluster. An increase in the values of $w_i$, $i = 1, \ldots, 4$ is required.

The results of the sensitivity analysis of responses Neur_M and Evcl_M to changes in components $z_i$, $i = 1, \ldots, 4$, during patient transitions between clusters indicated in Fig. 2 are shown in Figure 6 and in tables 15-17.

The joint sensitivity of the Euclidean metric and the neuron metric defined by the Euclidean – $E(N, E)$ in tables 10-12 defined by the Euclidean metric between the values of $N$ and $E$ in these tables – by degree of decrease to changes in components $z_i$ does not correspond to the order of significance of changes in the values of components $z_i : z_1, z_4, z_3, z_2$, which is due to the insignificant influence on the analysis results of the small values of parameters $w_1, w_2, w_3, w_4$ of the neuron metric (1) not proportional to the values of the Euclidean metric $E$ (2) and the greater influence of the values of (2).

The values of $N$ and $E$ in tables 6–8 are in the ranges $0 \leq N \leq 10$ and $0 \leq E \leq 100$. An increase in the given parameters $w_i$, $i = 1, \ldots, 4$ is proposed. Based on the analysis of

Table 13: Sensitivities of Neur_M and Evcl_M to changes in components $z_i$ during transition from the cluster with pathology to the cluster without pathology

| Parameter | $N$ | $E$ | $E(N, E)$ |
|---|---|---|---|
| $z_1$ | 100 | 44 | 109 |
| $z_2$ | 17 | 50 | 53 |
| $z_3$ | 33 | 47 | 57 |
| $z_4$ | 50 | 46 | 68 |

Table 14: Filled table with cluster centers and metric values (1) – (3) for $w_1 = -1, w_2 = 0.17, w_3 = 0.33, w4 = 0.5$

| $Z_1$ | $Z_2$ | $Z_3$ | $Z_4$ | Neur_M | Evcl_M | Pathology | Metric |
|---|---|---|---|---|---|---|---|
| 0.73 | 0.93 | 0.62 | 0.64 | 0.00 | 1.48 | + | -0.01 |
| 0.60 | 0.64 | 0.75 | 0.72 | 0.01 | 1.36 | ± | 0.02 |
| 0.80 | 0.84 | 0.94 | 0.92 | 0.01 | 1.75 | - | 0.02 |

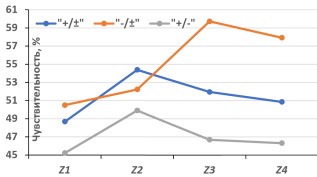

Figure 7: Results of sensitivity analysis of metrics Neur_M and Evcl_M to changes in the values of factors $z_{ij}$ during transitions between clusters indicated in Fig. 3.

Table 15: Sensitivities of Neur_M and Evcl_M to changes in components $z_i$ during transition from the cluster with pathology to borderline cases

| Parameter | $N$ | $E$ | $E(N, E)$ |
|---|---|---|---|
| $z_1$ | 10 | 48 | 49 |
| $z_2$ | 2 | 54 | 54 |
| $z_3$ | 3 | 52 | 52 |
| $z_4$ | 5 | 51 | 51 |

Table 16: Sensitivities of Neur_M and Evcl_M to changes in components $z_i$ during transition from the cluster without pathology to borderline cases

| Parameter | $N$ | $E$ | $E(N, E)$ |
|---|---|---|---|
| $z_1$ | 10 | 49 | 50 |
| $z_2$ | 2 | 52 | 52 |
| $z_3$ | 3 | 60 | 60 |
| $z_4$ | 5 | 58 | 58 |

Table 17: Sensitivities of Neur_M and Evcl_M to changes in components $z_i$ during transition from the cluster with pathology to the cluster without pathology

| Parameter | $N$ | $E$ | $E(N, E)$ |
|---|---|---|---|
| $z_1$ | 10 | 44 | 45 |
| $z_2$ | 2 | 50 | 50 |
| $z_3$ | 3 | 47 | 47 |
| $z_4$ | 5 | 46 | 46 |

cases 1-5, the values $w_i, \quad i = 1, \ldots, 4$, should satisfy conditions (4), (5) and be in the range

$$0 < w_i < 1, \quad i = 1, \ldots, 4. \tag{6}$$

At the same time, do not take values of $w_i$ close to 1 and 0 to ensure correct quantitative and qualitative analysis of data in clusters. Analysis of the graphs in Figures 3-5 allows us to identify similar behavior (parallel direction of movement) of 2 graphs of transition to the group with presence of pathology from the norm and from borderline cases and behavior different from this of the graph of movement from the norm to borderline cases of the presence of the disease, which allows programmatically (using computer analysis tools) to distinguish the presence of a group with pathology from the group of patients without it.

## 4 Sensitivity of cluster metrics to spinal changes during pathology development

### 4.1 Analysis of sensitivity of cluster metrics for $w_1 < 0, w_2 > 0, w_3 > 0, w_4 > 0$

Let for the weight coefficients of the Neuron growth metric (1), in addition to conditions (4), (5), the condition be satisfied

$$w_1 < 0, \quad w_2 > 0, \quad w_3 > 0, \quad w_4 > 0 \tag{6}$$

Figure 8 shows 20 cases of values of weight coefficients (1) selected according to (4)-(6).

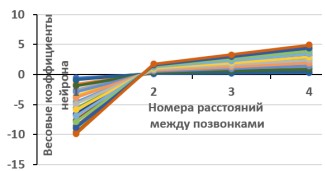

Figure 8: Selected cases of weight coefficients of the Neuron growth metric with $w_1 < 0, \quad w_2 > 0, \quad w_3 > 0, \quad w_4 > 0, \quad w_1 < w_2 < w_3 < w_4$

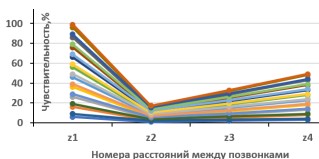

Figure 9: Sensitivity of neuron $N$ to changes in normalized distances $z_1, z_2, z_3, z_4$ for weight coefficients presented in Figure 7

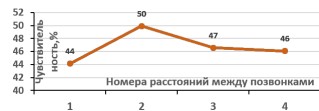

Figure 10: Sensitivity of Euclidean metric $E$ to changes in normalized distances $z_1, z_2, z_3, z_4$

### 4.2 Analysis of sensitivity of cluster metrics for $w_1 < 0, w_2 < 0, w_3 > 0, w_4 > 0$

Let for the weight coefficients of the Neuron growth metric (1), in addition to conditions (4), (5), the condition be satisfied:

$$w_1 < 0, \quad w_2 < 0, \quad w_3 > 0, \quad w_4 > 0 \tag{7}$$

Figure 7 shows 30 cases of values of weight coefficients (1) selected according to (4)-(6).

### 4.3 Analysis of sensitivity of cluster characteristics to spinal changes during pathology development

Figures 14-16(watch Appendix) present bar charts reflecting the presence of sensitivity in % of the neuron metric $N(1)$, Euclidean metric $E(2)$ and the Euclidean metric $E(N, E)$ from the values of metrics (1), (2) in relation to changes in intervertebral distances $z_1, z_2, z_3, z_4$.

Histograms were constructed separately for clusters 1 and 2 of weight coefficients of the neuron (1) presented in Figures 7, 11(watch Appendix) and for all selected weight coefficients from the 2 clusters.

According to Figures 15, 16, 17(watch Appendix), metrics (1), (2) are most sensitive (with probability 100%) to changes in the 2nd intervertebral distance $z_2$. The second in decreasing sensitivity (with probability 88%) of cluster metrics is the intervertebral distance $z_1$. The least sensitive (60%) is the intervertebral distance $z_3$.

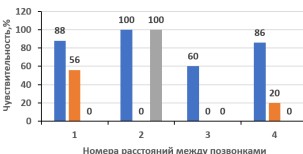

Figure 11: Sensitivity for all data ($w_1 < 0, \quad w_2 < 0$ or $w_2 > 0, \quad w_3 > 0, \quad w_4 > 0, \quad w_1 < w_2 < w_3 < w_4$) of Euclidean metric E (gray graph), neuron metric $N$ (orange bars) and Euclidean metric from the values of metrics $E(N, E)$ (blue bars) to changes in normalized intervertebral distances $z_1, z_2, z_3, z_4$

## 5 Conclusion

An analysis of the sensitivity of responses (1), (2) to changes in the values of intervertebral distances $z_1, z_2, z_3, z_4$ of patients during the transition from the cluster of "healthy" to the cluster "with pathology" was performed. Graphs of selected weight coefficients, graphs based on the results of sensitivity analysis of metrics (1), (2) and histograms of metrics for changes in intervertebral distances for two clusters separately and together were constructed. According to the study, the most probable cause of transition between the "healthy" and "diseased" clusters (Table 1) is a change in the second intervertebral distance.

Also, a 3rd cluster of weight coefficients will be considered in the future, satisfying the condition:
$$w_1 < 0, \quad w_2 < 0, \quad w_3 < 0, \quad w_4 > 0, \quad w_1 < w_2 < w_3 < w_4$$
and a total sample of 3 clusters for effective diagnosis of pathology depending on the selection of weight coefficients of the neuron growth. To confirm the hypothesis about the ranking of distances $z_2, z_1, z_4, z_3$ by sensitivity of responses (1), (2), further data analysis, in particular correlation analysis, is planned.

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

# A    Appendix

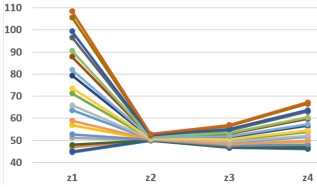

Figure 12: Values of the Euclidean metric $E(N, E)$ from the values of sensitivities $N$, $E$ presented in Figures 8, 9 for weight coefficients in Figure 7

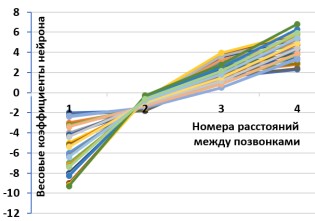

Figure 13: Weight coefficients with $w_1 < 0, \quad w_2 < 0, \quad w_3 > 0, \quad w_4 > 0, \quad w_1 < w_2 < w_3 < w_4$

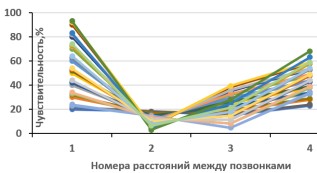

Figure 14: Sensitivity of neuron $N$ to changes in normalized distances $z_1, z_2, z_3, z_4$ for weight coefficients presented in Figure 11

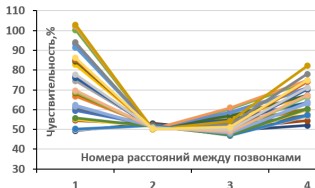

Figure 15: Values of the Euclidean metric $E(N, E)$ from the values of sensitivities $N$, $E$ presented in Figures 12, 9 for weight coefficients in Figure 11

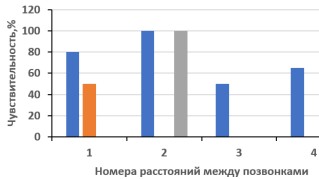

Figure 16: Sensitivity for cluster 1 (with $w_1 < 0$, $w_2 > 0$, $w_3 > 0$, $w_4 > 0$, $w_1 < w_2 < w_3 < w_4$) of Euclidean metric $E$ (gray bar), neuron metric $N$ (orange bars) and Euclidean metric from the values of metrics $E(N, E)$ (blue bars) to changes in normalized intervertebral distances $z_1, z_2, z_3, z_4$

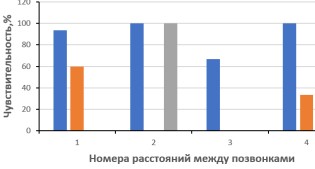

Figure 17: Sensitivity for cluster 2 (with $w_1 < 0$, $w_2 < 0$, $w_3 > 0$, $w_4 > 0$, $w_1 < w_2 < w_3 < w_4$) of Euclidean metric $E$ (gray bar), neuron metric $N$ (orange bars) and Euclidean metric from the values of metrics $E(N, E)$ (blue bars) to changes in normalized intervertebral distances $z_1, z_2, z_3, z_4$

