# OpenReview forum: "Methodology For The Comprehensive Study Of A Multidimensional Statistical Sample In The Diagnosis Of Spinal Diseases"
_mathai.club/MathAI/2026/Conference — MathAI 2026 Conference Submission_

### Official Review · Reviewer_L1s4 · 2026-03-10
**Weak accept for Life Sciences & Medicine track of MathAI-2026. Need fix Russian-English translation in figure legends and references.**

**Rating:** 7
**Confidence:** 4

**Review:**

Weak accept for Life Sciences & Medicine track.
Track recommendation: Life Sciences & Medicine (medical application: osteochondrosis diagnosis via lumbar X-ray distances z1-z4 clustering/sensitivity).
1. Mathematical Rigor is rather average.
"Neurogrowth" metric . Sensitivity as % change ratios across cluster transitions (pathology→borderline→healthy). No p-values, CIs, statistical tests, or validation. Cluster centers (Table 1) from prior k-means; no robustness analysis. Basic but adequate for medical stats.
No AI in the keywords, only statistical and medical keywords.
2. Novelty & Contribution - good.
Identifies z2 (2nd lumbar intervertebral distance, 100% sensitivity) > z1 (88%) > z3 (60%) as osteochondrosis diagnostic factors via weight enumeration (50 configs). 3-cluster interpretation (healthy/pathology/borderline) from X-ray measurements. Practical: prioritizes z2 for early detection protocols. Incremental vs. standard clustering [Yakimov 2011]; no ML comparison but actionable medical insight.
3. Relevance to MathAI Life Sciences & Medicine - excellent.
Strong domain fit: statistical methodology improves osteochondrosis diagnosis (lumbar spine degeneration affecting millions). Cluster sensitivity → clinical prioritization (focus X-ray precision on z2). Bridges stats/medicine; cites relevant works (Goldberg 2019 degeneration modeling). Less theoretical than core MathAI but valuable applied biomedicine.
4. Technical Quality is good.
Redundant figures (17 in total). All the formulas are shown.
Cluster centers realistic (healthy z2=14.3±0.8mm > pathology 9.1±1.3mm). Sensitivity tables consistent across cases (z1>z4>z3>z2 via E(N,E)); scale effects noted (large |w| → N>100%). No dataset size/N patients, clustering details (k-means params?), normalization method, inter-rater reliability for X-ray measurements. Figures support claims (histograms rank z2 highest). Sound medical stats, primitive by ML standards.
5. Clarity & Presentation: good
Clear pipeline: X-ray → z1-z4 → 3 clusters → sensitivity analysis → z2 prioritization. Tables excellent (Table 3: pathology→borderline transitions). Notation consistent.
Dense Russian-to-English translation issues ("neuron growth" unclear); repetitive cases (4 scales × transitions). No English translation for some references and figure legends.
Figures are referenced but quality unknown. Readable for clinicians.

---

### Official Review · Reviewer_ViYX · 2026-03-11
**Methodology For The Comprehensive Study Of A Multidimensional Statistical Sample In The Diagnosis Of Spinal Diseases**

**Rating:** 1
**Confidence:** 4

**Review:**

The paper does not propose a concrete algorithm or model. The so-called “multidimensional space” in the paper simply refers to four measurements of intervertebral distances (z1, z2, z3, z4). The authors introduce the Euclidean norm and a linear combination of these variables, which are then multiplied together without a clear theoretical justification. The weights of the linear combination appear to be chosen manually, and no optimization or learning procedure is described. In addition, the clustering mentioned in the paper is not explicitly formulated or implemented. The paper also lacks a proper review of related scientific work in this area.

---

### Decision · Program_Chairs · 2026-03-20

**Decision:**

Reject

**Comment:**

After careful evaluation by the Program Committee, we regret to inform you that your submission has not been accepted for presentation at MathAI 2026.

All submissions underwent a rigorous two-stage review process. Unfortunately, the reviewers identified one or more of the following concerns with your paper:

- Insufficient mathematical rigor or novelty relative to the existing body of work in the field;
- Presentation of results that substantially overlap with or rephrase previously published findings without clear original contribution;
- Significant issues with technical quality, including but not limited to broken or non-existent references, unsupported claims, or methodological gaps;
- Indications that the manuscript may have been generated with the assistance of large language models without substantial original intellectual contribution by the authors.

We received a large number of submissions this year, and the selection process was highly competitive. We encourage you to carefully consider the reviewers’ feedback (available through OpenReview), revise your work accordingly, and consider submitting an improved version to a future edition of MathAI or to another appropriate venue.

We appreciate your interest in MathAI and hope you will continue to engage with the conference community.

With kind regards,

MathAI 2026 Program Committee
International Conference on Mathematics of Artificial Intelligence
https://mathai.club
OpenReview: https://openreview.net/group?id=mathai.club/MathAI/2026/Conference
MathAI Telegram: https://t.me/MathAI_club
IAIC International AI Committee: https://t.me/iaic_world
Email: mathai.club@yandex.ru